# Structural insights into GABA~A~ receptor potentiation by Quaalude

Weronika Chojnacka[1,2], Jinfeng Teng[2], Jeong Joo Kim[3], Anders A. Jensen[4] & Ryan E. Hibbs [2,5] ✉

Methaqualone, a quinazolinone marketed commercially as Quaalude, is a central nervous system depressant that was used clinically as a sedative-hypnotic, then became a notorious recreational drug in the 1960s-80s. Due to its high abuse potential, medical use of methaqualone was eventually prohibited, yet it persists as a globally abused substance. Methaqualone principally targets GABA~A~ receptors, which are the major inhibitory neurotransmitter-gated ion channels in the brain. The restricted status and limited accessibility of methaqualone have contributed to its pharmacology being understudied. Here, we use cryo-EM to localize the GABA~A~ receptor binding sites of methaqualone and its more potent derivative, PPTQ, to the same intersubunit transmembrane sites targeted by the general anesthetics propofol and etomidate. Both methaqualone and PPTQ insert more deeply into subunit interfaces than the previously-characterized modulators. Binding of quinazolinones to this site results in widening of the extracellular half of the ion-conducting pore, following a trend among positive allosteric modulators in destabilizing the hydrophobic activation gate in the pore as a mechanism for receptor potentiation. These insights shed light on the underexplored pharmacology of quinazolinones and further elucidate the molecular mechanisms of allosteric GABA~A~ receptor modulation through transmembrane binding sites.

Methaqualone, commonly known as Quaalude, is a central nervous system (CNS) depressant that was prescribed in the 1960s–1980s as a sedative-hypnotic[1]. The drug promotes relaxation, calmness, drowsiness, and euphoria, and it was originally advertised as a safer alternative to barbiturates to treat insomnia[1]. Methaqualone was reported to induce deep sleep in patients with insomnia and to give rise to fewer side effects compared to barbiturates, including post-hypnotic drowsiness, fatigue, and headaches[2,3]. Methaqualone was also recognized as an effective anticonvulsant agent[4–6]. In addition to these positive attributes, however, methaqualone exhibited a high propensity for addiction and tolerance[7–9]. The euphoric and sedative-hypnotic effects of methaqualone led to its popularization as a recreational drug, often

consumed with alcohol, which increased its overdose potential. Due to widespread abuse, methaqualone was made illegal in 1984 by the Drug Enforcement Agency of the United States. While its access is now restricted in most of the world, methaqualone, referred to as Mandrax, remains a prevalent substance of abuse in South Africa[10,11]. Importantly, the popularization of methaqualone in the 1960s led to the synthesis of a range of related quinazolinones in clandestine laboratories, for example, mebroqualone[12], methylmethaqualone[13], SL-164[14], and more[15], resulting in recent overdoses[12,14]. Understanding the mechanism of action of methaqualone may contribute to the development of safer sedative and anticonvulsant therapeutics. Efforts to develop potent methaqualone derivatives that could serve as

[1]Biomedical Sciences Graduate Program, University of California San Diego, La Jolla, CA, USA. [2]Department of Neurobiology, University of California San Diego, La Jolla, CA, USA. [3]Protein Structure and Function, Loxo@Lilly, Louisville, CO, USA. [4]Department of Drug Design and Pharmacology, University of Copenhagen, Copenhagen, Denmark. [5]Department of Pharmacology, University of California San Diego, La Jolla, CA, USA. ✉e-mail: rehibbs@ucsd.edu

anticonvulsants[16–20] underscore the potential value of developing methaqualone derivatives as epilepsy therapeutics.

Methaqualone acts on the brain by selectively modulating type A γ-aminobutyric acid (GABA$_A$) receptors[21]. GABA$_A$ receptors belong to the Cys-loop superfamily of ligand-gated ion channels and are the major ionotropic inhibitory neurotransmitter receptors in the CNS[22]. In vivo, numerous GABA$_A$ receptor isoforms emerge from the 19 identified subunits[22], with a chloride-permeable ion channel being formed by the assembly of five identical or homologous subunits. The binding of the neurotransmitter GABA to β/α subunit interfaces in the extracellular domain (ECD) promotes the opening of the anion channel, which in most cases reduces neuronal excitability. Dysfunction of GABA$_A$ receptors leads to neurological disorders and mental illnesses including insomnia, anxiety disorders, amnesia, epilepsy, autism, depression, and schizophrenia[23–25]. GABA$_A$ receptors are targeted by many therapeutics and recreational drugs such as barbiturates, benzodiazepines, anticonvulsants, neurosteroids, anesthetics, and ethanol[26]. While the modes of action of some of these drugs are complex and comprise several activity components, their shared principal activity is positive allosteric modulation of GABA$_A$ receptors. They bind the receptor sites distinct from where GABA binds, and increase the GABA-induced response, thereby promoting nervous system depression. Despite methaqualone's dark history, it has been recognized as a promising molecule that could serve as a scaffold for novel modulators with sedative and anticonvulsant properties. Medicinal chemistry efforts identified a very potent methaqualone derivative, PPTQ (2-phenyl-3-(p-tolyl)-quinazolin-4(3H)-one)[27]. PPTQ displays ~50-fold higher modulatory potency (in terms of its EC$_{50}$ value) than methaqualone at the α1β2γ2 receptor subtype[21,27] and it acts as an ago-PAM, exhibiting agonist properties at concentrations ~300-fold higher than those mediating PAM activity[27] (Fig. 1a and Supplementary Fig. 1). Methaqualone when applied at very high concentrations (200–1000 μM), also exhibits minute but significant agonist activity[21] (Fig. 1a and Supplementary Fig. 1).

Although methaqualone and its derivatives are still being abused, their mechanisms of action remain understudied. Here we focus on structural and functional analyses of methaqualone and its more potent analog PPTQ at the GABA$_A$ receptor. We use single particle cryo-electron microscopy (cryo-EM) to first define binding sites and interactions for the two quinazolinones at the canonical synaptic α1β2γ2 GABA$_A$ receptor subtype. The structural results show binding sites that overlap with those for some general anesthetics and reveal a different pattern of interactions deeper at subunit interfaces. We use functional assays to test the importance of specific receptor residues as determinants for quinazolinone modulatory activity. Finally, we place our findings in the context of the previous structure-activity relationships exhibited by quinazolinone-based modulators and present structure-based mechanisms for quinazolinone action on GABA$_A$ receptors.

## Results and discussion
### Methaqualone and PPTQ share binding sites with general anesthetics
To elucidate the binding sites and molecular mechanisms for methaqualone and PPTQ, we purified a modified α1β2γ2 GABA$_A$ receptor and reconstituted it into saposin-lipid nanodiscs[28] in the presence of GABA and the respective quinazolinone (methaqualone or PPTQ), and collected cryo-EM datasets. Importantly, both the PAM and allosteric agonist activities of the two quinazolinones were preserved in this construct in which the structurally disordered intracellular domains from each subunit were removed (Fig. 1a and Supplementary Fig. 1). We used Fab fragments targeting the α1 subunits to facilitate particle alignment in cryo-EM data processing[29]. We obtained high-resolution cryo-EM structures for α1β2γ2 GABA$_A$ receptor complexes with GABA and methaqualone (2.8 Å) and with GABA and PPTQ (2.6 Å) (Methods,

Supplementary Table 1, Supplementary Fig. 2, Supplementary Movie 1, Supplementary Movie 4). Strong densities in the transmembrane domain allowed us to confidently position methaqualone and PPTQ at both β2/α1 subunit interfaces, loci that are well supported by mutagenesis experiments[21,27] (Fig. 1b, c for methaqualone and 1d, e, for PPTQ). The quinazolinone binding sites overlap with those of the general anesthetics etomidate and propofol (Fig. 2a, b), as well as with the lower affinity transmembrane binding sites of the anxiolytic diazepam[28,30], the new generation sedative-hypnotic zolpidem, and the convulsant DMCM[31]. Both methaqualone and PPTQ adopt equivalent poses at each of these two binding sites, and their common quinazolinone cores and tolyl rings superimpose well (Fig. 2b). The 2-methyl group in methaqualone (Fig. 2c) is replaced by a phenyl ring in PPTQ that orients intracellularly (Fig. 2d).

Several important modulators acting through the β/α TMD sites in the GABA$_A$ receptor leverage both common and distinct binding determinants. Most of these modulators interact with βN265, βM286, and αM236[21,32–35]. The carbonyl oxygen of both quinazolinones is positioned to form a hydrogen bond with the side chain amide of β2N265 (Fig. 2c, d). In addition, the carbonyl oxygen of this residue likely interacts with the 3-tolyl group of both drugs. This asparagine residue present in the β2/3 subunits corresponds to a serine in β1. While mutation of β2N265 to serine in the α6β2δ receptor subtype converts methaqualone from a PAM into a NAM, the reciprocal serine to asparagine mutation in β1 turns methaqualone from a NAM to a PAM at α6β1δ[21]. Mutation of β2N265 to methionine reduces the PAM activity of methaqualone and PAM and agonist activities of PPTQ significantly[21,27]. The β2/3N265M mutation also almost entirely eliminates potentiation by etomidate and significantly reduces potentiation of propofol[21,27,36,37] consistent with this bulkier residue sterically blocking the modulator binding site. In contrast to methaqualone, however, propofol and etomidate remain active at β1-containing receptors with serine in this position[38]. Thus, the β2/3-vs-β1 difference at this asparagine/serine position appears to contribute distinctly to the functional properties of these three modulators. This residue constitutes a PAM potency determinant for etomidate[38], acts as a key switch for methaqualone functionality[21], and is of little importance for propofol potency, but affects its efficacy[38]. These comparisons highlight the importance of subtle differences in modulator binding modes to the distinct functional profiles exhibited by propofol, etomidate, and methaqualone across the GABA$_A$ receptor subtypes. Receptor subtype-specific activities rooted in this difference are likely to contribute to their distinct in vivo effects.

The structures reveal both quinazolinones to be sandwiched between β2F289 and α1P233. While the 2-methyl group of methaqualone positions to form hydrophobic interactions with the phenyl of β2F289, the 2-phenyl ring of PPTQ forms π-stacking interactions with β2F289. β2M286 is positioned to form a π-sulfur interaction with the tolyl ring of methaqualone (Fig. 2c and Supplementary Movie 2). In the PPTQ-bound structure, the β2M286 sidechain slots between its tolyl and phenyl rings (Fig. 2d and Supplementary Movie 5). These additional hydrophobic interactions with β2M286 and β2F289 residues arising from PPTQ's 2-phenyl group likely play a role in its increased potency compared to methaqualone. The βM286W mutation is detrimental to the PAM and agonist activities exhibited by etomidate and propofol[33]. Similarly, mutation of β2M286 to either alanine or tryptophan completely abolishes PPTQ agonist activity and significantly decreases its PAM efficacy[27], whereas the β2M286W mutation causes a more subtle decrease in methaqualone's PAM activity[21]. Thus, both structural and functional results support that the higher modulatory potency displayed by PPTQ compared to methaqualone as an α1β2γ2 PAM may arise from a higher binding affinity of the modulator because of its increased hydrophobic interactions with β2M286 and β2F289.

One more hotspot for tuning modulator activity is at the α1M236 position, a residue on the complementary side of the interface forming

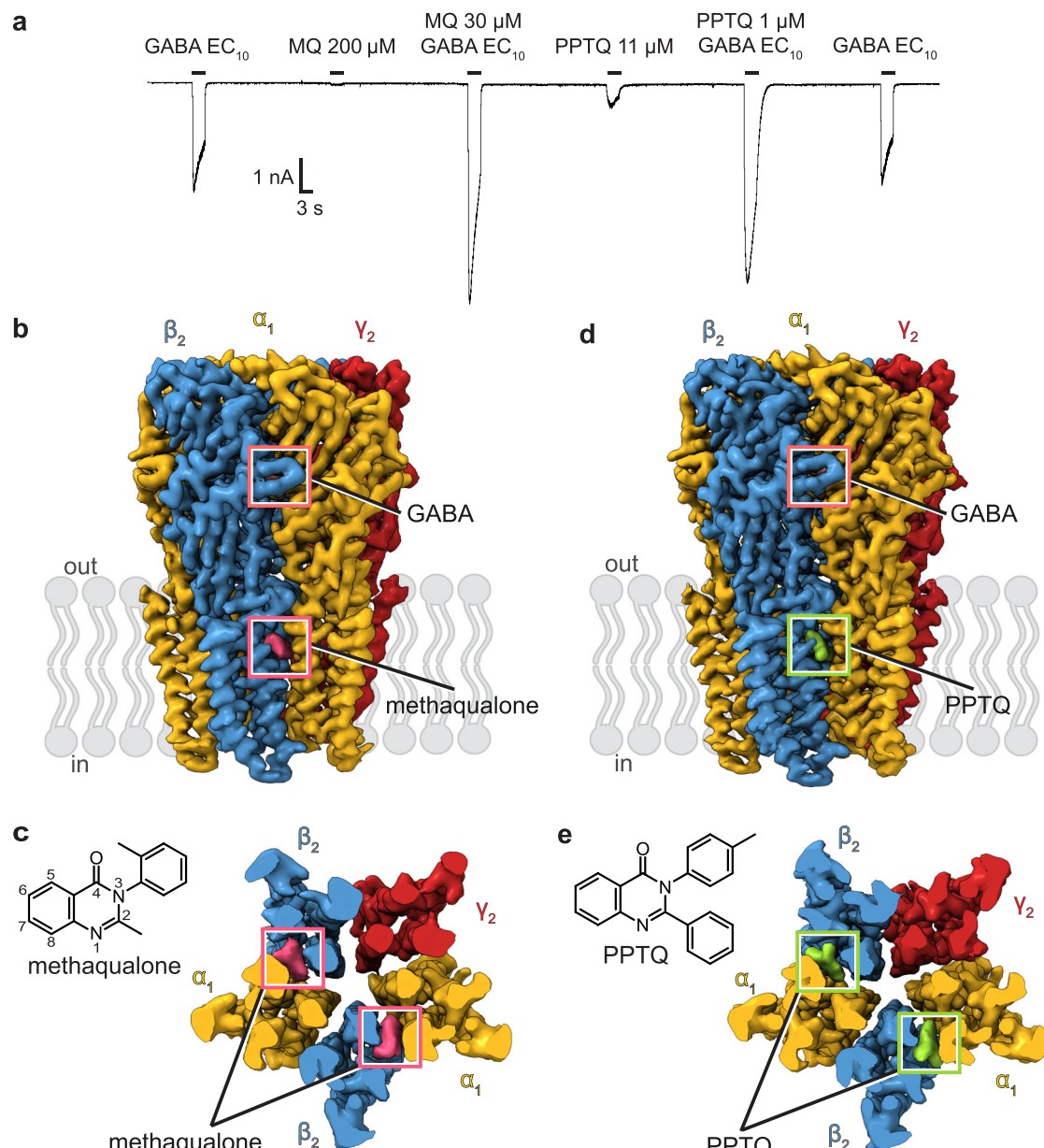

**Fig. 1 | Quinazolinones act as positive allosteric modulators through binding to TMD sites in the α1β2γ2 GABA_A receptor. a** Exemplary whole-cell patch-clamp electrophysiology trace showing the PAM and agonist activities displayed by methaqualone (MQ) and PPTQ on the EM receptor at their respective EC_50 concentrations; $n = 6$ recordings from independent cells; GABA EC_10 = 4 μM. **b** and **c** EM density map showing two methaqualone binding sites per receptor. **b** Side view of the receptor. **c** Cross section through the transmembrane domain and ligand chemical structure. **d** and **e** Same as **b** and **c** for PPTQ.

the floor of the quinazolinone site. Mutation of α1M236 to alanine has negligible effects on both the agonist and PAM activities of PPTQ[27], indicating that this methionine is not essential for modulator binding, whereas this alanine mutation increases etomidate modulatory efficacy[27]. Interestingly, an α1M236W mutation oppositely impacts PAM and agonist activities of methaqualone, PPTQ, and etomidate at the α1β2γ2 receptor[21,27,39,40]. Whereas all three modulators become dramatically more efficacious agonists at the α1M236W-containing receptor, their PAM efficacy plummets[21,27,39,40]. How do we understand these opposing effects on the different functional components of these ago-PAMs at the GABA_A receptor? The α1M236W mutation also increases receptor sensitivity to GABA[27,39,41], so the bulky tryptophan can be thought of as a covalent potentiator[39]. We hypothesize that there is a ceiling on PAM efficacy, and the increased baseline activity of the tryptophan mutant allows for a smaller fractional potentiation,

while at the same time, through lowering the energy barrier for activation, it increases the apparent activity of otherwise low-efficacy allosteric agonists acting through this site. This effect has been seen previously, where a gain-of-function tryptophan mutation in the Cys-loop at the ECD-TMD junction affected propofol PAM and agonist activities in opposite ways[42]. Effects of the abovementioned mutations on methaqualone and PPTQ activities are summarized in Supplementary Table 2.

The cryo-EM based mapping of the quinazolinone binding sites can be placed into the context of structure-activity relationship (SAR) studies of this drug class. Phenyl substitution at the 2-position of methaqualone (Fig. 1c) yields much more potent modulators, such as PPTQ and 2,3-diphenylquinazolin-4(*3H*)-one (PPQ)[27,43]. Here, we indeed find that the 2-phenyl in PPTQ engages in aromatic and hydrophobic stacking to logically stabilize its binding compared to

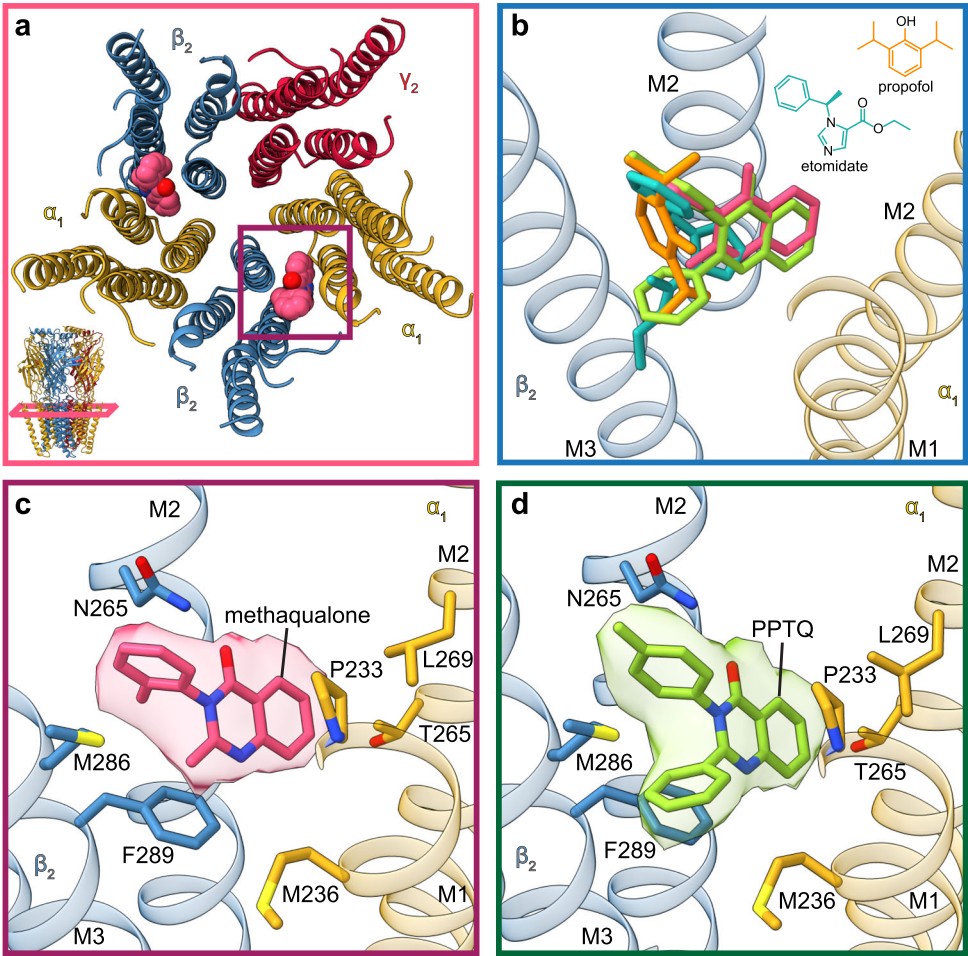

**Fig. 2 | Methaqualone binds to the same TMD β/α interface sites as the general anesthetics propofol and etomidate. a** Top view of the receptor TMD with one quinazolinone site boxed. **b** Comparison of quinazolinones and general anesthetics binding at the β2/α1 binding interface (orange – propofol, turquoise – etomidate, pink – methaqualone, green – PPTQ). **c** and **d** detailed view of methaqualone and PPTQ binding sites with residues shown by mutagenesis to affect quinazolinone activity as well as residues identified in our structures to likely interact directly with the ligands.

the parent methaqualone. The physicochemical properties and spatial orientation of this 2-phenyl group appear to be key for this gain in modulator potency as analogs comprising other aromatic or heteroaromatic rings as 2-substituents exhibit very low or no PAM activity[27]. These results are also supported by our structural findings as bulkier substituents than phenyl in the 2-position would likely cause a clash with neighboring residues or interfere with the lipid packing around the pocket. Polar features in this position are also unfavorable given the highly hydrophobic properties of this region of the binding site. The substitution pattern on the 3-phenyl ring of PPQ and PPTQ also appears to be important for its PAM activity[27]. For example, the ~10-fold higher PAM potency exhibited by Cl-PPQ compared to PPQ[27] can be rationalized by the fact that its *ortho*-chloro substituent in the 3-phenyl ring of Cl-PPQ seems to be well accommodated in the binding site where it may form additional electrostatic interactions with the carbonyl oxygen of α1I228. While the introduction of various substitutions in the 6-, 7- and 8-positions of the quinazolinone influences PAM potency distinctly, overall, substituents in these positions are not beneficial for GABA_A receptor modulatory activity[43]. These substituents would be present in the deepest part of the binding pocket where we found the quinazolinone core interacts with the α1 M2 helix (elaborated in the section below). The addition of bulky substituents would clash with both the principal and the complementary sides of the pocket, thereby lowering the PAM potency and/or efficacy. We expand on this structure-

oriented analysis of the quinazolinone SAR data in the Supplementary Discussion and Supplementary Fig. 3.

## Quinazolinones interact with the M2 helix of the complementary subunit

The GABA_A receptor TMD comprises four helices per subunit (M1-M4) with the five M2 helices forming a central ion conducting pore and gating the channel. There are two gates restricting ion flow, a desensitization gate at the −2′ amino acid position (counting residues from the bottom of the M2 helix) and an activation gate at and above the 9′ position[44–46]. While most of the modulators binding in the β/α interface pockets in the TMD occupy smaller, more superficial spaces, methaqualone and PPTQ insert deeply, making contact with the M2 helix of the complementary α1 subunit. The quinazolinones are thus positioned closer to the pore, enabling hydrophobic interactions with 10′ α1T265 (adjacent to the L9′ activation gate) and 14′ α1L269, both of which line the pore (Fig. 2c, d). To test whether these residues are involved in quinazolinone potentiation, we performed whole-cell patch-clamp electrophysiology assays on α1β2γ2 receptors comprising 10′ α1T265A and 14′ α1L269A mutants (in the cryo-EM construct background). To assess the impact of mutations on receptor activation by the neurotransmitter alone, we initially determined the GABA concentration-response relationships for the cryo-EM and mutant receptor constructs. We found that both mutations resulted in a modest gain-of-function effect for GABA-evoked receptor activation,

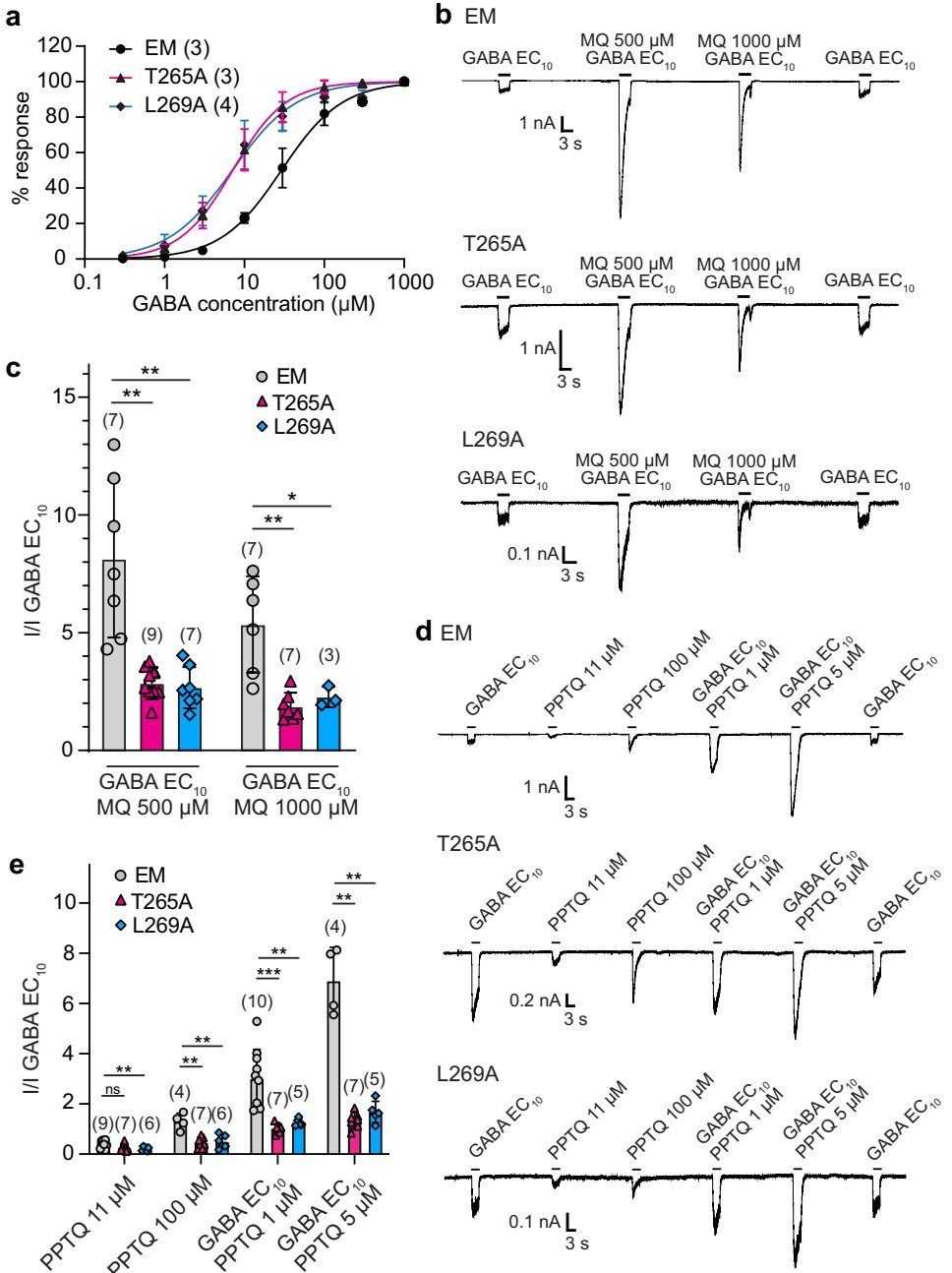

**Fig. 3 | Mutagenesis supports importance of deep binding sites for quinazolinones in the TMD β/α interface. a** Concentration-response curve showing GABA-mediated activation of α1T265A, α1L269A and EM α1β2γ2 receptors (EC$_{50}$ for EM; 7 μM for α1T265A and α1L269A). **b** Whole-cell patch-clamp electrophysiology traces showing activity of methaqualone (MQ) on mutated and EM receptors. **c** Bar graph showing methaqualone PAM activity on EM and mutated receptors normalized to GABA EC$_{10}$ = 4 μM for EM, 1 μM for α1T265A and α1L269A. **d** Whole-cell patch-clamp electrophysiology traces showing activity of PPTQ on mutated and EM receptors. **e** Bar graph for PPTQ PAM and agonist activities on EM and mutated receptors normalized to GABA EC$_{10}$. $n$ = recordings from independent cells. Results are shown as a mean response ± S.D.; $*p < 0.05$, $**p < 0.01$, $***p < 0.001$, $****p < 0.0001$.

the GABA EC$_{50}$ values being 29 μM for the cryo-EM construct and 7 μM for both mutants (Fig. 3a). As the M2 helices are central elements in the channel gating machinery, it is not surprising that mutations at these positions affect receptor activity. The gain of function in the 10′ α1T265A and 14′ α1L269A mutants may arise from removing steric restraints on M2 helix movement, thus decreasing the energy barrier to channel opening. We subsequently investigated the PAM activity of methaqualone on the mutated receptors by applying high concentrations of the modulator, 500 μM and 1000 μM, and observed that the modulatory efficacy for both concentrations decreased ~2-fold at both

mutants compared to the parent EM construct (Fig. 3b, c). Next, we tested PPTQ-mediated potentiation and activation abilities at the mutants. We observed that the efficacy of PPTQ both as an agonist and as a PAM were decreased in both mutants. The robust quinazolinone potentiation of GABA EC$_{10}$-evoked responses was dramatically diminished in the two mutant receptors (Fig. 3d, e). Moreover, direct activation by a high concentration of PPTQ was reduced significantly in both mutants, as well (Fig. 3d, e). These results indicate that the two M2 residues indeed constitute important determinants of both the potentiation and direct activation mediated by quinazolinones.

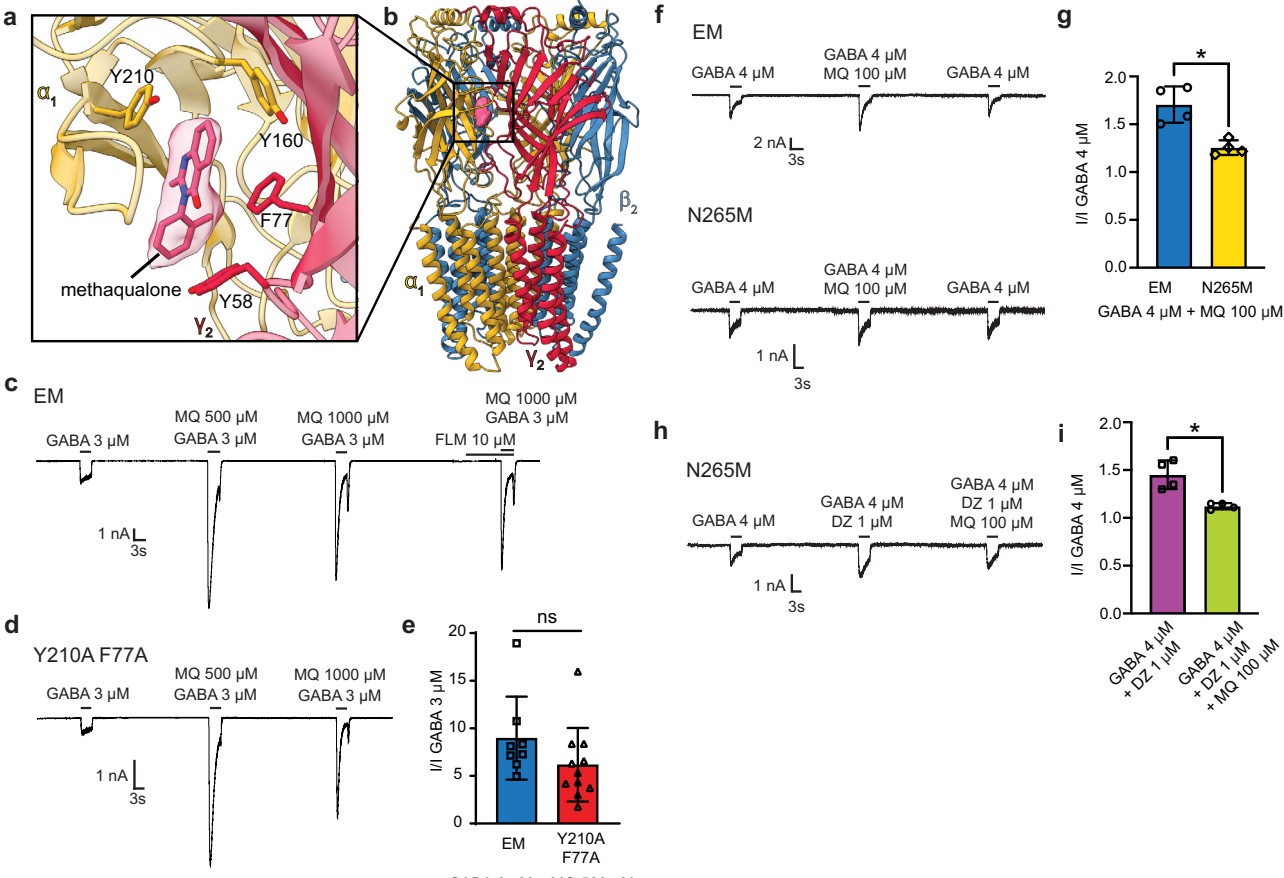

**Fig. 4 | The ECD benzodiazepine binding site is a silent site for methaqualone.**
**a** Close-up and **b** global views of the ECD α1/γ2 interface with methaqualone docked into the observed density. **c** Flumazenil competition and double mutant experiments, representative patch-clamp electrophysiology trace. **d** Representative patch-clamp electrophysiology trace for double mutant assay. **e** Statistical analysis of electrophysiology results comparing methaqualone responses in the EM and the double mutant receptors, the bar graph shows mean responses with standard deviation; $p = 0.17$, $n = 8$ recordings from independent cells for EM, $n = 11$ for Y210A/F77A. **f** Representative patch-clamp recording from

the test of methaqualone potentiation on the EM construct and its N265M mutant. **g** Statistical analysis of electrophysiology results comparing methaqualone potentiation on the EM construct and the N265M mutant, $p = 0.0119$, $n = 4$ recordings from independent cells. **h** Representative patch-clamp recording on N265M receptor from the competition assay for diazepam. **i** Statistical analysis of electrophysiology results comparing N265M receptors' response to diazepam without and with methaqualone, $p = 0.0194$, $n = 4$ recordings from independent cells. Results are shown as a mean response ± S.D. $p \leq 0.05$ was considered statistically significant, $*p < 0.05$, $**p < 0.01$, $***p < 0.001$, $****p < 0.0001$.

## Methaqualone-like density in the benzodiazepine site

Interestingly, the cryo-EM map for the methaqualone-bound structure reveals a strong density in the high-affinity benzodiazepine binding site located in the α1/γ2 interface at the ECD. Surprisingly, the density fits methaqualone well (Fig. 4a, b and Supplementary Movie 3), suggesting an additional, unpredicted binding site. A similar density is not observed in the map for the PPTQ-bound structure, and docking of PPTQ into this position in the methaqualone-bound structure results in significant steric clashes, suggesting that this putative binding site possesses selectivity among quinazolinones based on modulator size.

Hammer et al.[21] investigated the possible involvement of this site in methaqualone-mediated modulation. This study found that the α1β2γ2 receptor potentiation exerted by 300 μM methaqualone was not significantly affected by the presence of saturating concentrations of the benzodiazepine site antagonist flumazenil[26] or by the introduction of an α1H102R mutation, which is known to reduce the binding affinity of most modulators acting through this interface[47]. However, since methaqualone exhibits a bell-shaped concentration-response curve as an α1β2γ2 GABA_A receptor PAM, with a decreasing degree of potentiation observed at concentrations above 300 μM[21], we hypothesized that this ECD α1/γ2 pocket could be a low-affinity inhibitory site. We have previously observed such dual actions from the compound DMCM, which exerts NAM activity through the high-affinity

benzodiazepine site in the ECD, and PAM activity at higher concentrations through the same TMD site where methaqualone binds[31].

To probe the putative methaqualone binding to the ECD benzodiazepine site, we performed a series of experiments using whole-cell patch clamp electrophysiology. The cryo-EM density suggests that the drug would be sandwiched between α1Y210/γ2F77 in the ECD α1/γ2 interface (Fig. 4a, b and Supplementary Movie 3), and individual mutations of both residues have been shown to drastically decrease benzodiazepine activity[48,49]. Thus, we studied the modulation exerted by methaqualone at an α1Y210A/γ2F77A mutant receptor compared to its effects on the cryo-EM construct. We applied very high methaqualone concentrations, 500 μM, and 1000 μM, along with GABA EC_{10}, to HEK cells transiently expressing the receptors. The second test was a competition assay, where we measured the effects of GABA EC_{10}, 500 μM, and 1000 μM methaqualone together with and following a pre-application of 10 μM flumazenil on the cryo-EM construct. Consistent with previous findings[21], methaqualone-mediated modulation was affected neither by the presence of flumazenil nor by the introduction of the two mutations (Fig. 4c–e). These two assays invalidated our initial hypothesis that methaqualone acts as a low-affinity negative modulator through the benzodiazepine site, however, we remained curious whether methaqualone binding at this location had any measurable consequences on channel function. To answer this question,

we introduced a mutation, β2N265M, that was previously reported to be detrimental to methaqualone activity[21] and performed a competition assay against diazepam potentiation. We first tested the mutation's effect on methaqualone potentiation and observed that indeed most of the methaqualone activity was lost (Fig. 4f, g). We next compared GABA_A receptor potentiation by diazepam (1 μM) in the absence and presence of 100 μM methaqualone, the concentration used in cryo-EM, which gave rise to the methaqualone density in the benzodiazepine site (Fig. 4h, i). We found that while methaqualone had no measurable effect on receptor activation by GABA, it was able to block much of the PAM activity of diazepam.

Considering these results, we propose that methaqualone acts as a silent modulator (or competitive antagonist) at the α1/γ2 benzodiazepine site. The phenomenon of silent modulator binding at this locus has been seen before with α1β2γ2 GABA_A receptor complexes with other allosteric ligands, for example, flumazenil[26,28] and sulfated neurosteroids[50]. We present an alternative explanation for the biphasic modulatory profile of methaqualone, where at very high

concentrations, it becomes inhibitory. Our electrophysiology results reveal the occurrence of rebound currents following the washout of methaqualone (Figs. 3b, 4c), suggesting that at these higher concentrations, the PAM can also function as a pore blocker.

## Structural mechanism underlying potentiation by quinazolinones

The GABA_A receptor adopts three principal functional states: a conducting activated state and non-conducting resting and desensitized states. In both quinazolinone-bound structures the α1β2γ2 receptor adopts a desensitized-like state, similar to previous structures with other PAMs bound[28,30,31,50]. We observed that the desensitized-like structures with quinazolinones bound differ from the GABA-only structure (PDB ID: 6X3Z) primarily in the global contraction of the TMD and the width of the ion pore (Fig. 5a, b). Compared to both quinazolinone-bound structures, the GABA-only structure has a wider pore at the -2′ desensitization gate and a narrower pore at the level of the 9′ activation gate (near the midpoint of M2) (Fig. 5a, b). PAM

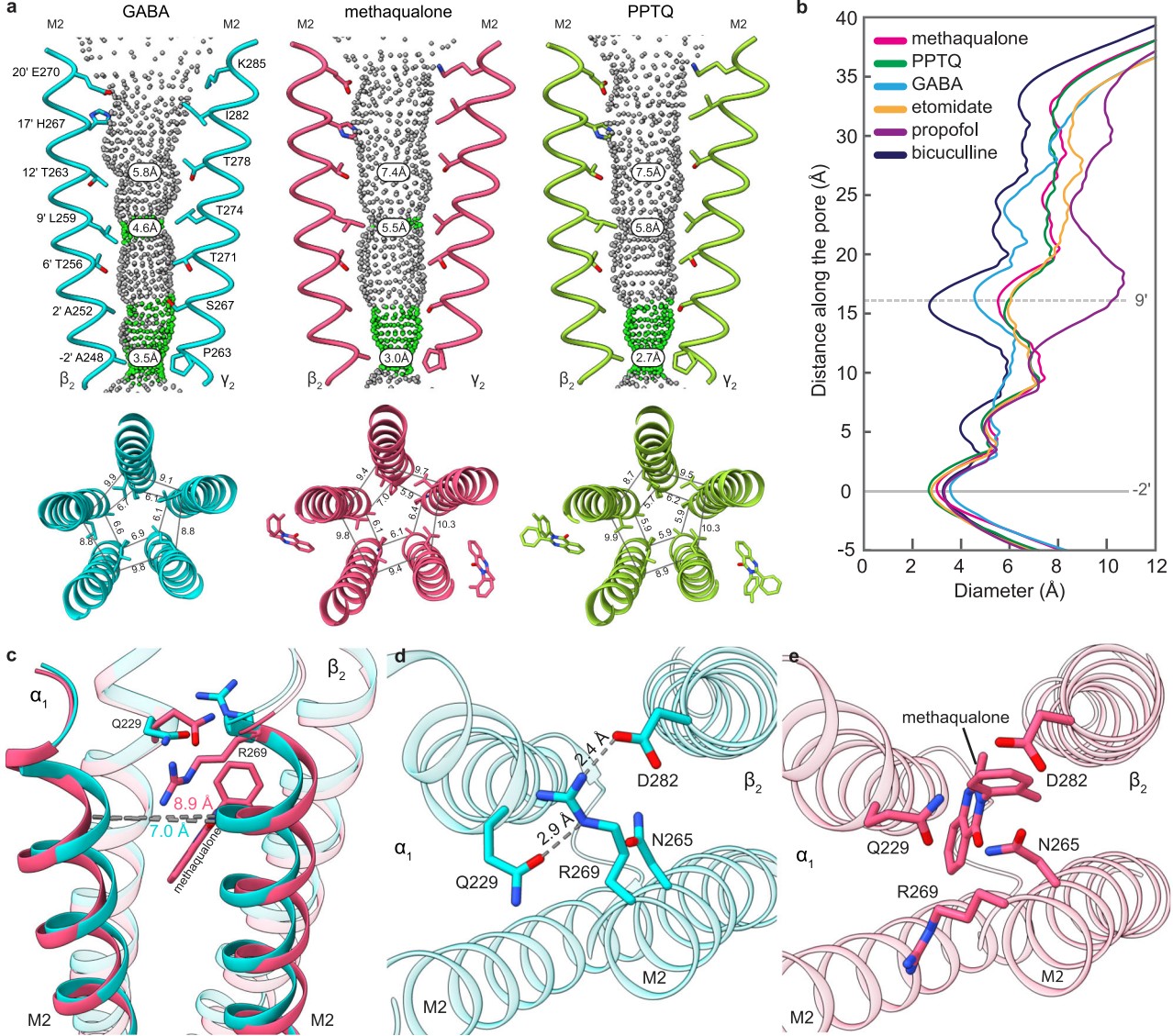

**Fig. 5 | Quinazolinone binding widens the pore above the 9′ gate. a** HOLE representation of the channel pore comparing resting and desensitization gates in GABA-only, methaqualone, and PPTQ structures. Pore diameters are indicated. **b** HOLE plot showing diameter along the pore for different structures (PDB IDs: GABA-6X3Z, etomidate-6X3V, propofol-6X3T, bicuculline-6X3S). **c** Comparison of

the M2 distances between GABA-only (turquoise) and methaqualone (pink) structures. **d** R269 (19′) position when the pocket is not occupied by a ligand (GABA-only structure, PDB ID: 6X3Z). **e** R269 (19′) position when the pocket is occupied by methaqualone.

binding stabilizes the separation of the M2 helices from the adjacent β and α subunits. Distances between the neighboring M2s (measured from Cα of α1S272 and β2T266) are 8.9 Å and 9 Å in methaqualone and PPTQ structures, respectively, versus 7 Å in the GABA-only structure (Fig. 5c). We propose that the quinazolinones primarily act through a mechanism similar to the other TMD-binding PAMs, where receptor potentiation is achieved by destabilizing the activation gate, thus lowering the energy barrier to activation.

We sought to understand how quinazolinone binding in the β/α interface pockets alters the positions of the neighboring helices and amino acid residues, and whether a conserved mechanism underlies how the binding of β/α TMD pocket modulators gives rise to this pore widening at the activation gate. Superposition of different receptor-PAM complexes reveals that the side chain of βR269 (commonly referred to as the 19′ arginine) located at the top of M2 orients distinctively as a function of β/α allosteric binding pocket occupancy. In the GABA-only structure (PDB ID: 6X3Z), the βR269 side chain orients toward the β/α binding pocket (Fig. 5d and Supplementary Fig. 4), where it is positioned to form an electrostatic interaction with βN282 and a hydrogen bond with αQ229 (Fig. 5d). In the quinazolinone-bound structures, however, βR269 resides between the neighboring M2 helices and orients into the pore. Thus, while βR269 does not directly interact with the drug, its binding induces an allosteric change, where the neighboring residues in the pocket rearrange to accommodate the drug in the site and thereby push the arginine residue away (Fig. 5e).

In comparing other PAM-bound structures, we found that βR269 flips out of the allosteric pocket to orient between M2 helices only for PAMs that bind at β/α TMD interfaces. For propofol (6X3T), etomidate (6X3V)[28], DMCM (8DD3), and zolpidem (8DD2)[31] that occupy the β/α binding pockets, βR269 points into the pore, between the two M2 helices, just like in quinazolinone-bound structures (Supplementary Fig. 4a), whereas structures for GABA (6X3Z), phenobarbital (6X3W), flumazenil (6X3U)[28], and neurosteroids (8SGO, 8SID, 8SI9, 8FOI, 8G5F)[50,51], where the β/α PAM pockets are empty, all have βR269 orienting toward the β/α pocket (Supplementary Fig. 4b). Finally, when comparing complexes where the ion channel is in a resting-like state, we observed another, distinct βR269 orientation that has been noted before[52]. In the receptor structures bound by the competitive antagonist bicuculline (6X3S, 6HUK)[28,30], and by the pore blocker picrotoxin (6HUG, 6X40, 6HUJ)[28,30], βR269 resides at the interface of M2 and M1 helices of the complementary subunits (Supplementary Fig. 4c). These observations suggest three distinct conformations of the arginine located at the top of the β/α allosteric binding site, and its spatial orientation seems to be linked with the receptor state and/or the modulator occupancy of this interface. The βR269 residue, which is conserved across all (except for ρ) GABAAR and α GlyR subunits, has been suggested to play an important role in conformational changes of the receptor during gating[52–57]. Although the 19′ arginine is highly conserved, this conformational change is unique to the β subunit and is observed only at the β/α TMD binding pocket. The architecture of the barbiturate binding pockets, located at the α/β and α/γ TMD interfaces, allows the arginine side chain to orient toward the PAM pocket, while PAM is bound, without creating a steric clash with neighboring amino acids.

In conclusion, we observe a trend among PAMs that bind to the TMD β/α binding pocket in the GABAA receptor. All PAMs contribute to widening the pore at and above the level of the 9′ activation gate, which would destabilize the resting state and thereby enhance the potentiation of GABA responses. At a finer level of detail, we mapped trends in conformational changes among studied modulators. β subunit 19′ arginine side chains at the β/α interface adopt three different orientations as a function of ligand binding: above the PAM binding pocket; between two neighboring M2 helices; and between the M2 and M1 helices of the complementary α subunit.

## Lipid interactions with the protein and quinazolinones

Based on cryo-EM map densities and information about the lipid configuration in the inner or outer leaflet of the cell membrane[58] we were able to position lipid molecules interacting with the receptor (Supplementary Fig. 5a, b for methaqualone, d,f for PPTQ). Both quinazolinone-bound structures show similar patterns of lipid interactions, but lipid densities are better resolved and there are two more lipids defined in the higher-resolution PPTQ map than in the methaqualone map. In the outer leaflet, we observed strong densities for two lipids adjacent to the β2/α1 TMD binding pockets positioned to make hydrophobic interactions with the protein and with the quinazolinone bound in the interface (Supplementary Fig. 5c for methaqualone; e,g for PPTQ). We modeled phosphatidylethanolamine (POPE) in these sites, as this is the most abundant lipid in the brain extract used in nanodisc reconstitution (~33%) and fits well into the experimental density. Two recent publications report the acyl chain of a lipid wedging into the β2/α1 TMD binding pockets where quinazolinones and general anesthetics bind[50,51]. In both quinazolinone-bound structures, the acyl chain of the POPE adjacent to α1 remains in close contact with the drug at approximate distances of 4-5 Å from the modulator. These two lipids at the β2/α1 TMD binding pockets adopt similar poses in structures with other β2/α1 TMD-binding ligands, including general anesthetics, picrotoxin[28], neurosteroids[50], zolpidem, and DMCM[31]. Interestingly, the lipid tail extending past the ligand is not visible in all structures. For structures with ECD-binding modulators like GABA, flumazenil, and bicuculline[28], the POPE adjacent to β2 is not well ordered, and for the POPE adjacent to α1, only the lipid head and shorter partial acyl chains are visible. These observations suggest that the β2/α1 TMD-binding drugs stabilize the bound lipids and that the lipids stabilize the bound drugs. In one of the PPTQ binding pockets (γβ/αβα), based on the experimental density, we identify a third ordered lipid: a POPC adjacent to the two POPEs (Supplementary Fig. 5e). These lipids at the β2/α1 interface were the only ones we observed in the outer leaflet.

We were able to identify well-ordered lipids located in the inner leaflet as well (Supplementary Fig. 5a, b for methaqualone, d,f for PPTQ). The distribution of ordered lipids in the quinazolinone-bound structures resembles the distribution in the GABAA receptor complex with pregnenolone sulfate, a negative modulator found to act as a pore blocker[50]. In the PPTQ-bound structure, we modeled a POPE molecule at one of the β2/α1 interfaces. The acyl chains of this POPE occupy the allopregnanolone binding site[50] adjacent to the M1 and M4 helices of the α1 subunit. For both quinazolinone-bound structures, we identified a phosphatidylserine (POPS) at the α1/β2 interface with its acyl chains adjacent to the β2 subunit. Another well-ordered POPS molecule is present at one of the α1/γ2 interfaces, interacting primarily with the α1 subunit.

In this study, we sought to understand how methaqualone family drugs act on GABAA receptors. Structure-function analysis of these compounds has lagged behind other GABA receptor modulators in part due to restrictions in their access to research. Here, we obtained two high-resolution structures of the α1β2γ2 GABAA receptor in complex with GABA and the sedative-hypnotic methaqualone, and with GABA and the more potent quinazolinone derivative PPTQ. The parent compound, methaqualone, is noteworthy in its sedative-hypnotic and anticonvulsant activities. We found that both quinazolinones bind in the same β2/α1 TMD interface pockets targeted by the general anesthetics etomidate and propofol. The quinazolinones bind deeper in this subunit interface than the previously characterized modulators, resulting in functionally important interactions with the M2 helix of the α1 subunit. The quinazolinones support a trend among PAMs binding to the β2/α1 TMD interfaces, where occupancy of the site stabilizes a separation of the principal and complementary M2 helices, weakening activation gate interactions to facilitate channel opening. Our structural results provide a 3D blueprint for the interpretation of existing

SAR data. The broader goal is to enable rational design of a new generation of quinazolinone analogs, toward limiting abuse potential while preserving desirable anti-convulsant and sedative-hypnotic components.

## Methods

### Receptor expression and purification

The α1β2γ2 GABA$_A$ receptor was expressed using a tri-cistronic construct as described previously[28,59]. Briefly, three genes corresponding to each subunit were placed in the pEZT-BM expression vector in the order of β2-γ2-α1. Genes were separated with a 22 amino acid long self-cleaving P2A peptide. Each subunit in the EM construct was modified by removing the M3-M4 loop and replacing it with a SQPARAA linker[29,60]. The N-terminus of the γ2 subunit was tagged with a twin-strep tag for affinity purification. For the PPTQ-GABA$_A$ receptor complex, Bacmam viral expression was used. Bacmam virus was produced in Sf9 cells and titered as previously described for the α4β2 nicotinic receptor[59]. HEK293S GnTI⁻ cells in suspension (total of 4.8 L), at a cell density of 3.5-4x10$^6$ cells/mL, were transduced with a multiplicity of infection (MOI) of 0.5 and the subunits were expressed for 72 h at 30 °C with 8% CO$_2$. To enhance expression, 3 mM sodium butyrate (Sigma Aldrich) was added during transduction.

For the methaqualone-GABA$_A$ receptor complex, a stable cell line was created using a Sleeping Beauty transposon system[61]. Adherent HEK293S GnTI⁻ cells were co-transfected with 1.9 µg pSBtet vector (pSBtet-GP, item #60495) carrying the EM construct and with 0.1 µg SB100X (pCMV(CAT)T7-SB100, item #34879) transposase carrying vector, using Lipofectamine2000 (Invitrogen) and the manufacturer's protocol. Twenty-four hours after transfection, cells were selected by incubation with 1 µg/mL puromycin. The selection was carried out until all cells showed fluorescence. Cells were then moved to a suspension culture. A total of 6.4 L of HEK293 GnTI⁻ cells, at density of 3.5-4x10$^6$ cells/mL, were then induced with 2 µg/mL of doxycycline and incubated, shaking, for 48 h at 30 °C with 8% CO$_2$, with addition of 3 mM sodium butyrate.

Cells were harvested by centrifugation and resuspended in 150 mM NaCl, 20 mM Tris, pH 7.4 (TBS buffer), 1 mM phenylmethanesulfonyl fluoride (PMSF; Sigma-Aldrich), 2 mM GABA (Sigma-Aldrich), and the target ligands: 100 µM 2-phenyl-3-(p-tolyl)quinazolin-4(3H)-one (PPTQ, Chembridge Corporation) or 100 µM methaqualone (obtained through the NIDA Drug Supply Program). Cells were mechanically lysed and centrifuged at 10,000 × $g$ for 20 mins. The resulting supernatant, containing cell membranes, was centrifuged at 186,000 × $g$ for 2 h. Resulting membrane pellets were homogenized using Dounce homogenizer, and solubilized for 1 h at 4 °C with nutating, in the TBS buffer enriched with 40 mM n-dodecyl-β-D-maltoside (DDM, Anatrace), 1 mM PMSF, 2 mM GABA, and corresponding ligands. Solubilized membranes were centrifuged for 40 min at 186,000 × $g$ and the supernatant was passed through the Strep-Tactin XT Superflow affinity resin (IBA-GmbH). The resin was washed with TBS buffer containing 2 mM DDM, 0.01% (w/v) porcine brain polar lipids (Avanti), 2 mM GABA and corresponding ligands. The protein was eluted using TBS buffer with 2 mM DDM, 0.01% (w/v) porcine brain polar lipids (Avanti), 2 mM GABA, corresponding ligands at 0.1 mM, and 50 mM biotin (Sigma-Aldrich).

### Nanodisc reconstitution

The plasmid for saposin A expression was obtained from Salipro Biotech AB. Reconstitution was conducted as previously described[28,31] using a modified protocol of Lyons et al.[62]. The concentrated receptors were mixed with porcine brain polar lipids (Avanti) and incubated at room temperature for 10 min. Subsequently, saposin was added and the mixture was incubated for 2 min. The mixture was prepared in 1:230:30 molar ratio of protein, lipids, and saposin. To initiate reconstitution, the solution was diluted ~10-fold with TBS buffer. Bio-Beads

SM-2 (Bio-Rad) at a concentration of 200 mg/mL were added to the solution to remove detergent. Then the mixture was rotated overnight at 4 °C, and then the Bio-Beads were removed the next day. The sample was collected for size-exclusion chromatography.

### Monoclonal antibody digestion and Fab purification

1F4 monoclonal antibody (mAb) against the α$_1$ subunit of the α1β2γ2 GABA$_A$ receptor (IgG2b, κ) was raised using standard methods (Monoclonal Core, Vaccine and Gene Therapy Institute, Oregon Health & Science University). Fab fragments were generated using papain cleavage of mAb at a final concentration of 1 mg/mL for 2 h at 37 °C in 50 mM NaPO$_4$, pH 7.0, 1 mM EDTA, 10 mM cysteine, and 1:30 (w/w) papain. 30 mM iodoacetamide was used to quench the reaction at 25 °C for 10 min. Fab fragments were purified by anion exchange using a HiTrap Q HP (GE Healthcare) column in 10 mM Tris, pH 8.0, and a NaCl gradient elution. Fab purified with this method was used for PPTQ-GABA$_A$ receptor complex.

### Recombinant Fab expression in Sf9 cells

The C-terminally 8xHis-tagged light and heavy chain genes for 1F4 Fab were subcloned into the pFastBac Dual vector (Thermofisher). Baculovirus was prepared using the Bac-to-Bac method (Thermofisher). The virus was amplified in Sf9 cells (ATCC CRL-1711) and used to infect a large Sf9 cell suspension culture at 27 °C. After 72 h expression, the media were collected, supplemented with a cOmplete™ EDTA-free protease inhibitor cocktail tablet (Roche), and sterile filtered. Two liters of media containing Fab protein were concentrated and dialyzed to 200 mL of TBS buffer pH 7.4 by ultrafiltration using a 50,000 cutoff VIVAFLOW 200 membrane (Sartorius). The sample was then applied onto a 5 mL HisTrap HP column (GE Healthcare). The 1F4 Fab was eluted with a linear imidazole gradient after several steps of washing (4 column volumes with 10, 20, 40, and 60 mM imidazole). The elution fractions were pooled and dialyzed against the TBS buffer to remove imidazole. Fab purified with this method was used for the methaqualone-GABA$_A$ receptor complex sample for cryo-EM.

### Cryo-EM sample preparation

The receptor-nanodisc complex was mixed with IF4 Fab fragment in a 3:1 (w/w) ratio and incubated on ice for 15 min. The mixture was concentrated to 500 µL and injected onto Superose 6 Increase 10/300 GL column (GE Healthcare) that was previously equilibrated with TBS supplemented with 2 mM GABA and respective ligand. Peak fractions were analyzed by fluorescence-detection size-exclusion chromatography, using tryptophan fluorescence. Single peak fractions were pooled and concentrated to 6-8 mg/mL (280 nm absorbance). To induce random orientation of the protein on a grid, 0.5 mM fluorinated Fos-Choline-8 (Anatrace) was added to the protein solution immediately before freezing grids. Grids were plunge frozen into liquid ethane using Vitrobot Mark IV (FEI). 3 µL of protein sample mixed with fluorinated Fos-Choline-8 was placed onto glow-discharged (PELCO easiGlow) for 80 seconds at 30 mA copper R1.2/1.3 200 mesh holey carbon grids (Quantifoil) before blotting for 3 s at 100% humidity and at the temperature of 4 °C.

### Cryo-EM data collection and processing

Cryo-EM data was collected at Pacific Northwest Center for Cryo-EM (PNCC) over 48 h on a 300 kV Titan Krios Microscope (FEI) equipped with K3 direct electron detector (Gatan) and a GIF quantum energy filter (20 eV) (Gatan) in a super-resolution mode. All datasets were processed using RELION 3.1[63], as follows: dose-fractionated images were gain normalized, 2x Fourier binned, aligned, dose-weighted, and summed using MotionCor2[64]. Gctf[65] was used to estimate the contrast transfer function (CTF). Particle picking was performed using crYOLO[66]. Picked particles were subjected to two rounds of 2D classification in RELION 3.1[63]. Full-size particles were extracted and 2D

classes that exhibited a clear GABA$_A$ receptor shape were picked for subsequent ab initio 3D model generation using 3000–5000 particles. Subsequent 3D classification was performed and all 3D classes exhibiting high-resolution features were picked for 3D refinement. Subsequently, polishing was performed. For methaqualone, two datasets were merged after polishing. CTF refinement was then done. Since the TMD of the gamma subunit is intrinsically disordered, focused classification without alignment was performed after subtracting the signal from the rest of the receptor. A final 3D refinement and postprocessing were performed next.

### Model building, refinement, and validation

The GABA$_A$ receptor complex with 1F4 Fab, GABA, and etomidate (PDB ID: 6X3V) was used as a starting point for modeling the PPTQ complex. Ligands were removed and the coordinates were docked into the PPTQ experimental map using UCSF Chimera[67]. The finalized PPTQ model was used as a starting model for the methaqualone complex. Manual adjustments and building were done in Coot[68]. The model was subjected to global real space and B-factor refinement with stereochemistry restraints in Phenix[69]. Geometry restraints for PPTQ and methaqualone were generated using PRODRG[70]. Model quality was assessed with Phenix and MolProbity[71]. Pore radius profiles were analyzed using Hole2[72]. Structural figures were made in UCSF ChimeraX[73]. Structural biology software packages were compiled by SBGrid[74].

### Electrophysiology

Whole-cell voltage-clamp recordings were collected on the adherent HEK293S GnTI⁻ that were transiently transfected with the tri-cistronic pEZT construct used for structural analyses, as well as with GFP protein in pEZT for selection. The amount of plasmids used for transfection was 0.2–0.6 μg, and transfection was performed according to Lipofectamine2000 manufacturer's protocol. The protein was expressed for 1–3 days at 30 °C. On the day of recording, cells were plated onto a 35 mm dish and washed with bath solution (140 mM NaCl, 2.4 mM KCl, 4 mM MgCl$_2$, 4 mM CaCl$_2$, 10 mM HEPES pH 7.3, and 10 mM glucose). Borosilicate pipettes were pulled and polished to an initial resistance of 2–4 MΩ. Pipettes were filled with the pipette solution (100 mM CsCl, 30 mM CsF, 10 mM NaCl, 10 mM EGTA, and 20 mM HEPES pH 7.3). Cells were clamped at −75 mV. The recordings were made using an Axopatch 200B amplifier, sampled at 5 kHz, and low pass filtered at 2 kHz using Digidata 1440 A (Molecular Devices), and analyzed with pClamp 10 software (Molecular Devices). Ligand solutions were prepared in a bath solution. A gravity-driven RSC-200 rapid solution changer (Bio-Logic) was used for solution exchange.

### Statistical analysis of electrophysiology data

Statistical analysis in Fig. 3 and Fig. 4 was performed using GraphPad Prism 9.2.0 software (GraphPad Software, Inc., La Jolla, CA). Data are expressed as means ± standard deviation of at least three recordings from independent cells. The two-tailed Welch's t-test was used. A p-value of ≤0.05 was considered statistically significant.

### Reporting summary

Further information on research design is available in the Nature Portfolio Reporting Summary linked to this article.

## Data availability

The data that support this study are available from the corresponding authors upon request. The cryo-EM maps have been deposited in the Electron Microscopy Data Bank (EMDB) under accession codes EMD-43485 (GABA + PPTQ). EMD-43475 (GABA + methaqualone). The atomic coordinates have been deposited in the Protein Data Bank (PDB) under accession codes 8VRN (GABA + PPTQ); 8VQY (GABA + methaqualone). Previously published structures compared in this study include: 6X3Z, 6X3T, 6X3V, 6X3W, 6X3U, 6X3S, 6HUK, 6HUG, 6X40, 6HUJ, 8DD3, 8DD2, 8SGO, 8SID, 8SI9, 8FOI, 8G5F. Source data are provided with this paper.

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

## Acknowledgements

We thank Colleen Noviello for screening the cryo-EM grids, Sean Burke, Huanhuan Li, Jia Zhou, Hao Jiang, and Colleen Noviello for feedback on the manuscript, and staff, especially Omar Davulcu, at the Pacific Northwest Cryo-EM Center for cryo-EM data collection. Single-particle cryo-EM grids were screened at the University of Texas Southwestern Medical Center Cryo-Electron Microscopy Facility, which is supported by the CPRIT Core Facility Support Award RP170644. A portion of this research was supported by NIH grant U24GM129547 and performed at the PNCC at OHSU and accessed through EMSL (grid.436923.9), a DOE Office of Science User Facility sponsored by the Office of Biological and Environmental Research. This project was supported by a Predoctoral Fellowship from the American Heart Association (24PRE1189840) to W.C. and a grant from the NIH (DAO47325) to R.E.H.

## Author contributions

W.C. performed the biochemistry, cryo-EM sample preparation, data processing, model building, refinement, structural analysis, electro-physiology, and created the figures and drafted the manuscript with R.E.H. J.T. performed electrophysiology. J.J.K. generated recombinant Fab used in the methaqualone complex structure determination. A.A.J. provided PPTQ and drafted structure-activity material. All authors were involved in the manuscript revision.

## Competing interests

The authors declare no competing interests.
