## [Peer Review File · Nature Communications]

Structural insights into GABAA receptor potentiation by QuaaludeReviewer #1 (Remarks to the Author):

The manuscript by Chojnacka et al. describes high-resolution structures of methaqualone (Quaalude) and its derivative PPTQ bound to the $\alpha 1\beta 2\gamma 2$ GABA-A receptor. The compounds are found in the anesthetic binding sites in the TMD at the b/a interfaces and in the high-affinity benzodiazepine site in the ECD at the a/g interface. The manuscript further describes and discusses the structural aspects of GABA-AR activation by the two compounds. Overall the study is expertly conducted, following numerous similar papers from the Hibbs' lab. The findings are of interest to researchers in the GABA-AR and anesthetic fields. My comments and criticism are generally minor, given below.

1. Methaqualone actions are indeed less studied, although unclear if that is, as the authors note, due to its restricted status (l. 74). I suggest the authors present stronger rationale for conducting this study. Related to this, I wish the authors expanded on how they believe this study contributes to development of safer sedatives and anticonvulsants as well as treatment of addiction and overdose (l. 49-50, 339-341). Specifically and particularly for the latter, I wonder if there is anything beyond the immediate, but of course varying, functional effects that distinguishes the numerous compounds that bind to the b/a interfacial anesthetic site.

2. l. 142-149, the effects of M236W. Increased constitutive activity likely underlies the findings. The functional mechanism of this was described in detail in PMID: 29439087 that the authors may wish to consult.

3. Electrophysiological experiments, Figure 1a. It may be preferable to use the same concentration of MQ (or PPTQ) in direct activation vs. potentiation experiments. Due to a concurrent inhibitory effect at higher concentrations the response to 200 μ M MQ200 may be misleadingly small. The potential non-functional binding in the ECD is very interesting. Please test and show if methaqualone inhibits the potentiating actions of benzos acting through this site.

4. Minor issues. l. 55, I would recommend specifying "ionotropic" as there are also major metabotropic inhibitory neurotransmitter receptors in the CNS. l. 63, probably all these drugs have concurrent NAM actions, some minor rewriting would be helpful here. l. 72, very high and very weak are not informative. Rephrase. l. 234, silent modulator (oxymoron?). Did the authors mean a competitive antagonist? Fig. 4b, I would find it useful if there was, besides the Angstrom scale, a scale corresponding to TM2 positions. Suppl. Fig. 1, ordinate label, perhaps delete "relative".

Reviewer #2 (Remarks to the Author):

The Hibbs lab presents cryo EM structures of $\alpha 1\beta 2\gamma 2$ with two quinazolinones present in upper TMD and ECD pockets. They derive a mechanism of action which is supported by mutational and electrophysiological studies and comparisons with other ligated structures. Overall, the manuscript is well written and the findings well interpreted, add value to the field and should be published with some minor improvements as detailed:

1. IUPHAR recommendation is to reserve subscripts for stoichiometry, and explicitly suggests to not use subscripts for subunit isoforms. It is suggested that this is adhered to.
2. Line 58: "anion channel" is preferred over "chloride channel" to account for the bicarbonate permeability
3. Line 215: The H-R difference does not eliminate benzodiazepine binding, it rather interferes with binding of specific substances, but not e.g. for flumazenil – this should be reworded. One convention used by many is the term "diazepam insensitivity" for $\alpha 4$ and $\alpha 6$ and the H-R conversion mutants.
4. The reader would be helped with a table (or supplementary table) summarizing the mutational studies that were compiled, and in part performed in this study. Perhaps it can somehow be integrated to Figure 2.

5. Panel b of Figure 2 might benefit from the use of a different rendering style for sidechains compared to the ligands.

Reviewer #3 (Remarks to the Author):

In the current study, Chojnacka et al investigate the mechanisms by which Quinazolinones interact and modulate ionotropic GABAA receptors. They use single particle cryo-EM to determine structures of a hetero-trimeric GABAA receptors in a lipid environment, in complex with GABA and 2 different Quinazolinone derivatives. The structures identify specific drug binding sites from which the authors are able to make inferences/predictions on how chemical derivations of Quinazolinones might affect their affinities and they also reveal key information underlying how sequence specific diversity in various subtypes of GABAA receptors affect drug action. The authors all perform a variety of electrophysiological experiments to complement their structural observations and also infer mechanisms underlying the allosteric modulation by the drugs. The reconstructions are of high-quality and the paper is well-written. However, I think the paper would become even better with greater clarification and a more careful consideration of some mechanistic explanations.

Line 69: Please clarify what is meant by: "... is ~50-fold more potent...". Is this referring to affinity/efficacy?

Line 93: Does Fab affect the modulation (efficacy or affinity) of methaqualone or PPTQ?

Line 134: I'm not sure how the structure supports "...higher potency of PPTQ...". From the structure it seems that the pi-stacking interactions of PPTQ could lead to a higher affinity, the functional consequences of a more "effective" (or tighter) interaction is not clear (it's not clear what the authors mean by "potency")

Line 146-149: While I understand what the authors are speculating, I think it would be extremely beneficial to provide an energy level diagram (in the SI may be). I agree that there could be a "ceiling on PAM efficacy" but it would depend on the specific scheme/mechanism of modulation by the PAM, for example: altering the intrinsic dynamics (pore opening) of the receptor vs increasing the GABA binding affinity. It could help to may be include specific citations to studies which show evidence for or explore such mechanisms of action for the PAMs under investigation/discussion.

Line 152: "Potent" = "high affinity" modulator?

Fig 3b/Supplementary Fig. 4: In the representative currents with 1 mM MQ and EC10 GABA, after current activation/desensitization there is a very quick "tail-like" current upon ligand wash-off. Is this a solution perfusion artifact or does MQ, at such high conc., also have pore blocking effects and the tail-like currents reflect some kind of fast-unblocking? (reminiscent of the action of open-channel blockers in many LGICs/VGICs)

Fig 4 and Line 254: I don't completely agree with this explanation. A wider separation at the 2' activation gate does not inform/imply energy barriers to activation. If anything, a wider pore might have a more direct correlation with ion selectivity or conductance rather than energy barriers to channel opening. An additional concern with this interpretation is that the ionic current traces also show enhanced desensitization in the presence of the drugs (~100 uM or higher concentrations) and structurally the desensitization gate, in presence of MQ/PPTQ, appears to be more constricted than that in presence of GABA alone almost to the same extent as the activation gate relaxes. So it's not clear why the authors choose to focus more on a mechanism of "potentiation" and sort of disregard the other aspects of drug action on structure/function. While I appreciate that activation and desensitization are inherently coupled processes, the explanation offered here to correlate the

potentiation activity of the drugs (observed over short 3-5 s applications) to the structures (obtained under conditions where receptors are stewing in the drugs for >24 hrs) is not particularly satisfying. I would encourage the authors to "discuss" these details with more careful considerations of different aspects of gating/allosteric mechanisms and may be cut back a little on making "conclusions".

Response to reviewers:

Reviewer # 1 comments:

The manuscript by Chojnacka et al. describes high-resolution structures of methaqualone (Quaalude) and its derivative PPTQ bound to the $\alpha 1\beta 2\gamma 2$ GABA-A receptor. The compounds are found in the anesthetic binding sites in the TMD at the β/α interfaces and in the high-affinity benzodiazepine site in the ECD at the α/γ interface. The manuscript further describes and discusses the structural aspects of GABA-AR activation by the two compounds. Overall the study is expertly conducted, following numerous similar papers from the Hibbs' lab. The findings are of interest to researchers in the GABA-AR and anesthetic fields. My comments and criticism are generally minor, given below.

We thank the reviewer for their enthusiasm about the interest of the study, and for their constructive critiques that we address below. These comments have helped us to strengthen the study.

1. Methaqualone actions are indeed less studied, although unclear if that is, as the authors note, due to its restricted status (l. 74). I suggest the authors present stronger rationale for conducting this study. Related to this, I wish the authors expanded on how they believe this study contributes to development of safer sedatives and anticonvulsants as well as treatment of addiction and overdose (l. 49-50, 339-341). Specifically and particularly for the latter, I wonder if there is anything beyond the immediate, but of course varying, functional effects that distinguishes the numerous compounds that bind to the β/α interfacial anesthetic site.

The reviewer makes a good point that we cannot be certain why there has been relatively little research done on quinazolinones compared to, for example, intravenous anesthetics and benzodiazepines. We speculate that restricted access contributed to lowering the interest in this compound in the scientific community due to it being difficult to access as a schedule 1 compound. Notably, methaqualone abuse remains and we believe that it is an understudied compound. We correct this part in the manuscript; line 76: *“Although methaqualone and its derivatives are still being abused, their mechanisms of action remain understudied.”* We removed the emphasis on the reason why. We also toned down this connection in the abstract. For rationale, there are potential analogies with other infamous drugs, for example THC, ketamine, and psychedelics like LSD and psilocybin. There has been a need for rigorous research on these drugs to test claims of therapeutic benefits, but restrictions on obtaining and studying these drugs essentially halted progress in understanding their mechanisms and potential value. Research on these drugs has, relatively recently, been undertaken, and the results appear, at least to us, very exciting and worthwhile. We suggest that there is a similar need to revisit quinazolinones. We (the Hibbs lab) were motivated to conduct this research after being contacted by a synthetic chemistry lab (Jensen lab) to provide a structural foundation for next steps in rationally improving selectivity and activity profiles of new derivatives.

Since there is a tremendous chemical diversity among compounds that bind at the TMD β/α interface, we believe that detailed insights into the mode of action of this class of allosteric modulators is expanding our knowledge about differences in binding modes compared to different modulators binding to the same sites, which is reflected in our study. The diverse modulators have a spectrum of activities from PAMs to NAMs, and varying selectivity. Both the selectivity and efficacy will contribute to how circuit activity is affected.

We agree with the reviewer that claiming that our results specifically will help address overdose and addiction might be too farfetched. We still believe that they might greatly contribute to development of safer quinazolinone-based therapeutics and there is indeed interest in developing such agents¹⁻⁵. Our study provides structural information of binding mode of this group of compounds. In the revised manuscript we state on lines 50-51: *“Understanding the mechanism of action of methaqualone may contribute to the development of safer sedative and anticonvulsant therapeutics.”*

2. I. 142-149, the effects of M236W. Increased constitutive activity likely underlies the findings. The functional mechanism of this was described in detail in PMID: 29439087 that the authors may wish to consult.

Thank you very much for drawing our attention to this study. We agree - this appears to be the same phenomenon we observed, from a mutant in another region, but the same conceptual mechanism. We have included this study and brief discussion thereof in the revised manuscript.

3. Electrophysiological experiments, Figure 1a. It may be preferable to use the same concentration of MQ (or PPTQ) in direct activation vs. potentiation experiments. Due to a concurrent inhibitory effect at higher concentrations the response to 200 μM MQ200 may be misleadingly small.

We thank the reviewer for this suggestion. As presented in the Introduction to the manuscript, the functional properties of both methaqualone and PPTQ as both agonists (direct activation) and PAMs have been characterized in detail in previous studies^{6,7}. Our use of the different methaqualone and PPTQ concentrations in Fig. 1a is to demonstrate the pharmacological differences between them as agonists and PAMs. Thus, the data in Figure 1a shows that the properties and potencies of the respective two modulators are mirrored in the EM construct in the patch-clamp recordings, which also lays the foundation for the mutagenesis studies in this work.

Our choice of 200 μM was based on the maximal potentiation of methaqualone observed in the previous study⁶. The example trace from their publication is shown below.

Even though 200 μM is the maximal potentiation concentration, we cannot rule out that there is some concurrent inhibition, however, at lower concentrations there is no significant direct activation seen for methaqualone – here it acts as a pure PAM. So, methaqualone is a low-potency allosteric agonist at the receptor, where high concentrations of the drug are needed to induce direct activation, however, at these high concentrations the inhibitory component of the modulator may also kick in.

The potential non-functional binding in the ECD is very interesting. Please test and show if methaqualone inhibits the potentiating actions of benzos acting through this site.

We thank the reviewer for this comment and agree that such an experiment would strengthen the conclusions we made in the study. Following the reviewers suggestion, we performed an experiment on a β_2N265M mutant that blocks most of the methaqualone activity⁶. We incorporated the results into the Supplementary Fig. 4e-h. Indeed, we found that MQ can inhibit diazepam potentiation. Accordingly, we now include MQ bound in the ECD site in the atomic model.

4. Minor issues.

I. 55, I would recommend specifying "ionotropic" as there are also major metabotropic inhibitory neurotransmitter receptors in the CNS.

Thank you. We addressed this in line 55: *"GABA_A receptors belong to the Cys-loop superfamily of ligand-gated ion channels and are the major ionotropic inhibitory neurotransmitter receptors in the CNS."*

I. 63, probably all these drugs have concurrent NAM actions, some minor rewriting would be helpful here.

We certainly agree with the reviewer that the actions of at least some of the modulators listed in this sentence are very complex and hold activity components as NAMs at GABA_A receptors as well as modulatory activities at other receptor classes. We believe that an elaborate account of these other activities would be out of scope of this Introduction. However, in the revised manuscript we have rephrased this sentence to; lines 64-65: *"While modes of action of some of these drugs are complex and comprise several activity components, their shared principal activity is positive allosteric modulation of GABA_A receptors. They bind the receptor sites distinct from where GABA binds, and increase the GABA-induced response, thereby promoting nervous system depression."*

I. 72, very high and very weak are not informative. Rephrase.

We agree with the reviewer comment and have rephrased the sentence in the revised manuscript; line 73: *"Methaqualone, when applied at very high concentrations (200-1000 μM), also exhibits minute but significant agonist activity."*

I. 234, silent modulator (oxymoron?). Did the authors mean a competitive antagonist?

We agree with the reviewer that the term “silent (allosteric) modulator” is paradoxical. Nevertheless, the term is used for ligands that bind to an allosteric site, but neither act as a PAM nor a NAM, i.e. is “silent”. Flumazenil is a classical example of a silent allosteric modulator acting through the benzodiazepine-binding site in the GABA_A receptors. The term “competitive antagonist” would be confusing, we think, as this term is typically used for antagonists targeting the orthosteric site, in this case the GABA-binding sites in the receptor.

To make this clearer to the reader we rewrote this section of the manuscript, lines 247-253.

Fig. 4b, I would find it useful if there was, besides the Angstrom scale, a scale corresponding to TM2 positions.

We thank for this suggestion, we added the scale; this certainly adds more clarity to the figure.

Suppl. Fig. 1, ordinate label, perhaps delete "relative".

Fixed.

Reviewer #2 comments:

The Hibbs lab presents cryo EM structures of $\alpha 1\beta 2\gamma 2$ with two quinazolinones present in upper TMD and ECD pockets. They derive a mechanism of action which is supported by mutational and electrophysiological studies and comparisons with other ligated structures. Overall, the manuscript is well written and the findings well interpreted, add value to the field and should be published with some minor improvements as detailed:

We thank the reviewer for enthusiasm and constructive comments about the manuscript.

1. IUPHAR recommendation is to reserve subscripts for stoichiometry, and explicitly suggests to not use subscripts for subunit isoforms. It is suggested that this is adhered to.

Fixed.

2. Line 58: “anion channel” is preferred over “chloride channel” to account for the bicarbonate permeability

Fixed.

3. Line 215: The H-R difference does not eliminate benzodiazepine binding, it rather interferes with binding of specific substances, but not e.g. for flumazenil – this should be reworded. One convention used by many is the term “diazepam insensitivity” for $\alpha 4$ and $\alpha 6$ and the H-R conversion mutants.

We agree with the reviewer. In the revised manuscript we have rephrased the sentence to; lines 220-221: “... or by the introduction of a $\alpha 1H102R$ mutation, which is known to reduce the binding affinity of most modulators acting through this interface.” Since the H-R mutation has been shown to inhibit the binding affinities or modulatory potencies of both 1,4- and 1,5-benzodiazepines as well as of several non-benzodiazepines like the Z-drugs (e.g. zolpidem) acting through this site, we think that this rephrased sentence is accurate.

4. The reader would be helped with a table (or supplementary table) summarizing the mutational studies that were compiled, and in part performed in this study. Perhaps it can somehow be integrated to Figure 2.

We understand that the reviewer is suggesting to present a new table to summarize the mutations described in sections “*Methaqualone and PPTQ share binding sites with general anesthetics*” and “*Quinazolinones interact with the M2 helix of the complementary subunit*”. We attempted to create a table summarizing effects of the mutations of the key pocket residues on methaqualone and PPTQ PAM and agonist activities (presented below). We are not convinced that such a table adds clarity to the manuscript. Our plan is to keep it out of the final manuscript, however, we are flexible and open to reviewers’ and editorial feedback.

Mutation	Methaqualone	PPTQ
$\alpha 1\beta 2^{M286W}\gamma 2$	Decrease in PAM activity.	Elimination of direct activation and pronounced decrease in PAM activity.
$\alpha 1\beta 2^{M286A}\gamma 2$	Not tested.	Elimination of direct activation and decrease in PAM activity.
$\alpha 1^{M236W}\beta 2\gamma 2$	Pronounced increase in direct activation.	Pronounced increase in direct activation and decrease in PAM activity.
$\alpha 1^{M236A}\beta 2\gamma 2$	Not tested.	No significant difference in both direct activation and PAM activity.
$\alpha 6\beta 2^{N265S}\delta$	Elimination of PAM activity.	Not tested.
$\alpha 6\beta 1^{S265N}\delta$	Rescued PAM activity.	Not tested.
$\alpha 1\beta 2^{N265M}\gamma 2$	Almost eliminated PAM activity.	Elimination of both direct activation and PAM activity.
$\alpha 1^{T265A}\beta 2\gamma 2$	Decreased PAM activity.*	Decreased PAM and agonist activities.*
$\alpha 1^{L269A}\beta 2\gamma 2$	Decreased PAM activity.*	Decreased PAM and agonist activities.*

*This study

5. Panel b of Figure 2 might benefit from the use of a different rendering style for sidechains compared to the ligands.

We agree that this panel was crowded. We addressed this in the revision by removing unnecessary details and now only focus on the positions of different drugs in the binding pocket.

Reviewer #3 Comments:

In the current study, Chojnacka et al investigate the mechanisms by which Quinazolinones interact and modulate ionotropic GABAA receptors. They use single particle cryo-EM to determine structures of a heterotrimeric GABAA receptors in a lipid environment, in complex with GABA and 2 different Quinazolinone derivatives. The structures identify specific drug binding sites from which the authors are able to make inferences/predictions on how chemical derivations of Quinazolinones might affect their affinities and they also reveal key information underlying how sequence specific diversity in various subtypes of GABAA receptors affect drug action. The authors all perform a variety of electrophysiological experiments to complement their structural observations and also infer mechanisms underlying the allosteric modulation by the drugs. The reconstructions are of high-quality and the paper is well-written. However, I think the paper would become even better with greater clarification and a more careful consideration of some mechanistic explanations.

We thank the reviewer for their careful reading of the study and very helpful questions and suggestions, which we respond to below.

Line 69: Please clarify what is meant by: "... is ~50-fold more potent...". Is this referring to affinity/efficacy?

This refers to the potencies displayed by PPTQ and methaqualone as PAMs, and it is based on comparisons between their EC_{50} values as PAMs in functional assays. So the statement is neither referring to binding affinity (even though the potency of a PAM in a functional assay is rooted in its binding affinity) nor to efficacy. We use the term potency in this way throughout the whole manuscript.

To make this clearer to the reader we have rephrased the sentence to; line 70: "*PPTQ displays ~50 higher modulatory potency (in terms of its EC_{50} value) than methaqualone at the $\alpha 1\beta 2\gamma 2$ receptor subtype...*" in the revised manuscript.

Line 93: Does Fab affect the modulation (efficacy or affinity) of methaqualone or PPTQ?

We have shown that this Fab is a weak positive allosteric modulator itself, shifting the EC_{50} for GABA by 2.5-fold to the left. For reference, see Zhu S et al., 2018⁸, Supplementary Figure 2d. In that same study, Fab was found to have no significant effect on benzo-site ligand flumazenil's activity or binding affinity. We have however not directly tested the effect of Fab on MQ or PPTQ binding or activity. If the reviewer/editor think this is a critical experiment, we can do it. We do not think any possible outcomes would alter our interpretations. The Fab is used as a tool for particle alignment and it has been shown to not substantially alter the ability of the receptor to undergo relevant conformational changes. The structural models allowed us to generate hypotheses that were then tested in functional assays in the absence of Fab, and in all cases the functional results and structural interpretations are internally consistent.

Line 134: I'm not sure how the structure supports "...higher potency of PPTQ...". From the structure it seems that the pi-stacking interactions of PPTQ could lead to a higher affinity, the functional consequences of a more "effective" (or tighter) interaction is not clear (it's not clear what the authors mean by "potency").

Analogously to our response to the reviewer comment about Line 69, here we also refer to the different modulatory potencies displayed by PPTQ and methaqualone at the $\alpha 1\beta 2\gamma 2$ GABA_A receptor, as assessed by their respective EC_{50} values as PAMs in functional assays. Exactly as the reviewer states, we propose that the pi-stacking interactions formed by PPTQ may yield higher binding affinity of this modulator, which is reflected in its higher modulatory potency compared to methaqualone.

To make this clearer to the reader we have rephrased the sentence to; line 137: "*Thus, both structural and functional results support that the higher modulatory potency displayed by PPTQ compared to methaqualone as a $\alpha 1\beta 2\gamma 2$ PAM may arise from a higher binding affinity of the modulator because of its increased hydrophobic interactions with $\beta 2M286$ and $\beta 2F289$.*" in the revised manuscript.

Line 146-149: While I understand what the authors are speculating, I think it would be extremely beneficial to provide an energy level diagram (in the SI may be). I agree that there could be a "ceiling on PAM efficacy" but it would depend on the specific scheme/mechanism of modulation by the PAM, for example: altering the intrinsic dynamics (pore opening) of the receptor vs increasing the GABA binding affinity. It could help to may be include specific citations to studies which show evidence for or explore such mechanisms of action for the PAMs under investigation/discussion.

We appreciate the conceptual suggestion. We spent some time sketching out energy diagrams but have been cofounded by the dimensions needed. Apo receptor, agonist alone, PAM alone at a low (PAM concentration), PAM alone at a high concentration (allosteric agonist), agonist + PAM at low and high concentration, all in the background of WT vs. mutant. It would be overwhelmingly complicated, and we do not

have experiments that directly test the energy levels. We feel it is intuitively logical that the balance between allosteric activation and allosteric potentiation of a modulator will be displaced towards the former in the M236W mutant, and that this conversely means that the “PAM component window” is reduced. We welcome further suggestions but for now we do not understand, after some consideration, how the suggested SI figure would add to the description in the text, beyond visualizing a hypothesis.

Line 152: “Potent” = “high affinity” modulator?

As outlined in some of our responses above, we prefer to use the term “potency” of the modulators rather than “affinity”, as we based the structure-activity relationship of the modulator series on functional data, and not on binding assay data.

Fig 3b/Supplementary Fig. 4: In the representative currents with 1 mM MQ and EC10 GABA, after current activation/desensitization there is a very quick “tail-like” current upon ligand wash-off. Is this a solution perfusion artifact or does MQ, at such high conc., also have pore blocking effects and the tail-like currents reflect some kind of fast-unblocking? (reminiscent of the action of open-channel blockers in many LGICs/VGICs)

We agree with the reviewer’s interpretation, that the tail current suggests channel block. Additionally, the biphasic concentration-response curve that was seen for methaqualone at both $\alpha 1\beta 2\gamma 2$ and $\alpha 6\beta 2\delta$ in Hammer et al.⁶ also suggests a lower-affinity inhibitory/block site. This low-affinity block site has also been proposed for other modulators binding to β/α TMD interface^{9–13}. We comment on this potential mechanism beginning on line 253.

Fig 4 and Line 254: I don’t completely agree with this explanation. A wider separation at the 2’ activation gate does not inform/imply energy barriers to activation. If anything, a wider pore might have a more direct correlation with ion selectivity or conductance rather than energy barriers to channel opening. An additional concern with this interpretation is that the ionic current traces also show enhanced desensitization in the presence of the drugs (~100 uM or higher concentrations) and structurally the desensitization gate, in presence of MQ/PPTQ, appears to be more constricted than that in presence of GABA alone almost to the same extent as the activation gate relaxes. So it’s not clear why the authors choose to focus more on a mechanism of “potentiation” and sort of disregard the other aspects of drug action on structure/function. While I appreciate that activation and desensitization are inherently coupled processes, the explanation offered here to correlate the potentiation activity of the drugs (observed over short 3-5 s applications) to the structures (obtained under conditions where receptors are stewing in the drugs for >24 hrs) is not particularly satisfying. I would encourage the authors to “discuss” these details with more careful considerations of different aspects of gating/allosteric mechanisms and may be cut back a little on making “conclusions”.

Thank you for this suggestion. We agree that making conclusive inferences based solely on structural comparisons is risky. We focused heavily on changes at the activation gate (9’) as it has been shown that its rotation is a hallmark of channel activation¹⁴. The literature also shows that 9’ polar mutants cause profound gain of function effects across the pentameric ligand gated channel superfamily¹⁵, by disrupting gate formation. Accordingly, a PAM that increases pore diameter, and thereby likelihood of hydration at 9’, could logically act through that structural disruption. Most allosteric modulators binding in GABA_AR TMD follow this trend. At the same time, we are unsure whether the stronger constriction at -2’ contributes significantly to channel activation/deactivation or the rate of desensitization. However, you raise a great point, in that the desensitization gate is closed, and closed more tightly, and this may counteract the destabilization of the activation gate; the mechanism may be more complex than we concluded. We did not directly measure the desensitization kinetics, however, what we observe is that the higher the GABA concentration, the faster the desensitization. Comparing whole-cell patch clamp recordings we do not see a notable change in desensitization rate by addition of methaqualone.

We have adjusted our language here to be more speculative in tone; line 269: *“We propose that the quinazolinones primarily act through a mechanism similar to the other TMD-binding PAMs, where receptor potentiation is achieved by destabilizing the activation gate, thus lowering the energy barrier to activation.”*

References

- (1) Ahmad, I.; Akand, S. R.; Shaikh, M.; Pawara, R.; Manjula, S. N.; Patel, H. Synthesis, Molecular Modelling Study of the Methaqualone Analogues as Anti-Convulsant Agent with Improved Cognition Activity and Minimized Neurotoxicity. *Journal of Molecular Structure* **2022**, *1251*, 131972. <https://doi.org/10.1016/j.molstruc.2021.131972>.
- (2) Patel, H. M.; Noolvi, M. N.; Shirkhedkar, A. A.; Kulkarni, A. D.; Pardeshi, C. V.; Surana, S. J. Anti-Convulsant Potential of Quinazolinones. *RSC Advances* **2016**, *6* (50), 44435–44455. <https://doi.org/10.1039/c6ra01284a>.
- (3) El-Azab, A. S.; ElTahir, K. E. H.; Attia, S. M. Synthesis and Anticonvulsant Evaluation of Some Novel 4(3H)-Quinazolinones. *Monatshefte für Chemie* **2011**, No. 142, 837–848. <https://doi.org/10.1007/s00706-011-0525-3>.
- (4) Zayed, M. F.; Ihmaid, S. K.; Ahmed, H. E. A.; El-adl, K.; Asiri, A. M.; Omar, A. M. Synthesis, Modelling, and Anticonvulsant Studies of New Quinazolines Showing Three Highly Active Compounds with Low Toxicity and High Affinity to the GABA-A Receptor. *Molecules* **2017**, *22* (188), 1–15. <https://doi.org/10.3390/molecules22020188>.
- (5) Zayed, M. F.; Ahmed, H. E. A.; Omar, A. S. M.; Abdelrahim, A. S.; El-Adl, K. Design, Synthesis, and Biological Evaluation Studies of Novel Quinazolinone Derivatives as Anticonvulsant Agents. *Medicinal Chemistry Research* **2013**, *22* (12), 5823–5831. <https://doi.org/10.1007/s00044-013-0569-5>.
- (6) Hammer, H.; Bader, B. M.; Ehnert, C.; Bundgaard, C.; Bunch, L.; Hoestgaard-Jensen, K.; Schroeder, O. H. U.; Bastlund, J. F.; Gramowski-Voß, A.; Jensen, A. A. A Multifaceted GABAA Receptor Modulator: Functional Properties and Mechanism of Action of the Sedative-Hypnotic and Recreational Drug Methaqualone (Quaalude). *Molecular Pharmacology* **2015**, *88* (2), 401–420. <https://doi.org/10.1124/mol.115.099291>.
- (7) Madjroh, N.; Rie, E.; Bundgaard, C.; Cecilia, P.; Jensen, A. A. Functional Properties and Mechanism of Action of PPTQ , an Allosteric Agonist and Low Nanomolar Positive Allosteric Modulator at GABA A Receptors. *Biochemical Pharmacology* **2018**, *147*, 153–169. <https://doi.org/10.1016/j.bcp.2017.11.006>.
- (8) Zhu, S.; Noviello, C. M.; Teng, J.; Walsh, R. M.; Kim, J. J.; Hibbs, R. E. Structure of a Human Synaptic GABAA Receptor. *Nature* **2018**, *559* (7712), 67–88. <https://doi.org/10.1038/s41586-018-0255-3>.
- (9) Khom, S.; Baburin, I.; Timin, E.; Hohaus, A.; Trauner, G.; Kopp, B.; Hering, S. Valerenic Acid Potentiates and Inhibits GABAA Receptors: Molecular Mechanism and Subunit Specificity. *Neuropharmacology* **2007**, *53* (1), 178–187. <https://doi.org/10.1016/j.neuropharm.2007.04.018>.
- (10) Hill-Venning, C.; Belelli, D.; Peters, J. A.; Lambert, J. J. Subunit-Dependent Interaction of the General Anaesthetic Etomidate with the γ -Aminobutyric Acid Type A Receptor. *British Journal of Pharmacology* **1997**, *120* (5), 749–756. <https://doi.org/10.1038/sj.bjp.0700927>.
- (11) Wooltorton, J. R. A.; Moss, S. J.; Smart, T. G. Pharmacological and Physiological Characterization of Murine Homomeric B3 GABA_A Receptors. *Eur J of Neuroscience* **1997**, *9* (11), 2225–2235. <https://doi.org/10.1111/j.1460-9568.1997.tb01641.x>.
- (12) Halliwell, R. F.; Thomas, P.; Patten, D.; James, C. H.; Martinez-Torres, A.; Miledi, R.; Smart, T. G. Subunit-selective Modulation of GABA Receptors by the Non-steroidal Anti-inflammatory Agent, Mefenamic Acid. *Eur J of Neuroscience* **1999**, *11* (8), 2897–2905. <https://doi.org/10.1046/j.1460-9568.1999.00709.x>.
- (13) Feng, H. J.; Macdonald, R. L. Proton Modulation of A1b3d GABAA Receptor Channel Gating and Desensitization. *Journal of Neurophysiology* **2004**, *92*, 1577–1585. <https://doi.org/10.1152/jn.00285.2004>.
- (14) Dämgen, M. A.; Biggin, P. C. A Refined Open State of the Glycine Receptor Obtained via Molecular Dynamics Simulations. *Structure* **2020**, *28* (1), 130-139.e2. <https://doi.org/10.1016/j.str.2019.10.019>.
- (15) Chang, Y.; Weiss, D. S. Allosteric Activation Mechanism of the A1 β 2 γ 2 γ -Aminobutyric Acid Type A Receptor Revealed by Mutation of the Conserved M2 Leucine. *Biophysical Journal* **1999**, *77* (5), 2542–2551. [https://doi.org/10.1016/S0006-3495\(99\)77089-X](https://doi.org/10.1016/S0006-3495(99)77089-X).

Reviewer #1 (Remarks to the Author):

The authors have nicely revised the paper. I have one general comment and one request for revision:

1. The comment is about competitive antagonism. There is no reason to limit the term to compounds binding to the orthosteric site. Competitive antagonism conveys mechanism.
2. Line 155. "...agonist activities in a similar way⁴²". The sentence is a bit confusing. Similar way here may refer to the previous sentence that described activation and potentiation being affected in opposite way. Alternatively, "similar way" may refer to propofol PAM and agonist activities within the same sentence, in which case it would, erroneously, state that the mutation increased both potentiation and activation. To be clear, gain of function mutations, or exposure to a background agonist, enhance the ability of a weak agonist to directly activate the receptor but reduce its apparent ability to potentiate activation by a third agonist. Perhaps replacing "similar way" with "in different directions" would make the point clear.

Reviewer #2 (Remarks to the Author):

This revision addresses all of my previously stated concerns. The table that is now in the reply letter, in my opinion, would be helpful as a supplementary item.

Looking forward to the published version!

We thank the reviewers for their careful reading, enthusiasm about the findings, and thoughtful feedback. Their suggestions made the study stronger and clearer.

Reviewer #1 (Remarks to the Author)

The authors have nicely revised the paper. I have one general comment and one request for revision:

1. The comment is about competitive antagonism. There is no reason to limit the term to compounds binding to the orthosteric site. Competitive antagonism conveys mechanism.

We agree and have updated the silent agonist terminology to also refer to MQ at the benzo site as a competitive antagonist.

2. Line 155. "...agonist activities in a similar way⁴²". The sentence is a bit confusing. Similar way here may refer to the previous sentence that described activation and potentiation being affected in opposite way. Alternatively, "similar way" may refer to propofol PAM and agonist activities within the same sentence, in which case it would, erroneously, state that the mutation increased both potentiation and activation. To be clear, gain of function mutations, or exposure to a background agonist, enhance the ability of a weak agonist to directly activate the receptor but reduce its apparent ability to potentiate activation by a third agonist. Perhaps replacing "similar way" with "in different directions" would make the point clear.

We thank the reviewer for their kind words. We agree with the request and address it in the final manuscript: "This effect has been seen previously, where a gain-of-function tryptophan mutation in the Cys-loop at the ECD-TMD junction affected propofol PAM and agonist activities in opposite ways."

Reviewer #2 (Remarks to the Author)

This revision addresses all of my previously stated concerns. The table that is now in the reply letter, in my opinion, would be helpful as a supplementary item.
Looking forward to the published version!

We thank the reviewer for their enthusiasm. We include the table in the Supplementary Information as Supplementary Table 2.